# Are Hamsters a Suitable Model for Evaluating the Immunogenicity of RBD-Based Anti-COVID-19 Subunit Vaccines?

**DOI:** 10.3390/v14051060

**Published:** 2022-05-16

**Authors:** Iuliia A. Merkuleva, Dmitry N. Shcherbakov, Mariya B. Borgoyakova, Anastasiya A. Isaeva, Valentina S. Nesmeyanova, Natalia V. Volkova, Vazirbek S. Aripov, Daniil V. Shanshin, Larisa I. Karpenko, Svetlana V. Belenkaya, Elena I. Kazachinskaia, Ekaterina A. Volosnikova, Tatiana I. Esina, Alexandr A. Sergeev, Kseniia A. Titova, Yulia V. Konyakhina, Anna V. Zaykovskaya, Oleg V. Pyankov, Evgeniia A. Kolosova, Olesya E. Viktorina, Arseniya A. Shelemba, Andrey P. Rudometov, Alexander A. Ilyichev

**Affiliations:** 1State Research Center of Virology and Biotechnology “Vector”, Rospotrebnadzor, World-Class Genomic Research Center for Biological Safety and Technological Independence, Federal Scientific and Technical Program on the Development of Genetic Technologies, 630559 Koltsovo, Russia; j.a.merkulyeva@gmail.com (I.A.M.); borgoyakova_mb@vector.nsc.ru (M.B.B.); anastasya_isaeva_1993@mail.ru (A.A.I.); nesmeyanova_vs@vector.nsc.ru (V.S.N.); volkova_nv@vector.nsc.ru (N.V.V.); aripov_vs@vector.nsc.ru (V.S.A.); shanshin_dv@vector.nsc.ru (D.V.S.); lkarpenko@ngs.ru (L.I.K.); belenkaya.sveta@gmail.com (S.V.B.); alenakaz@vector.nsc.ru (E.I.K.); volosnikova_ea@vector.nsc.ru (E.A.V.); esina_ti@vector.nsc.ru (T.I.E.); sergeev_ala@vector.nsc.ru (A.A.S.); titova_ka@vector.nsc.ru (K.A.T.); iuliakonyahina@yandex.ru (Y.V.K.); zaykovskaya_av@vector.nsc.ru (A.V.Z.); pyankov_ov@vector.nsc.ru (O.V.P.); kurchanovaea@gmail.com (E.A.K.); rudometov_ap@vector.nsc.ru (A.P.R.); ilyichev@vector.nsc.ru (A.A.I.); 2Russian-American Anti-Cancer Center, Altai State University, 656049 Barnaul, Russia; olesia_viktorina@mail.ru; 3Federal Research Center of Fundamental and Translational Medicine, 630060 Novosibirsk, Russia; arseniya.shelemba@mail.ru

**Keywords:** SARS-CoV-2, COVID-19, receptor-binding domain, animal models

## Abstract

Currently, SARS-CoV-2 spike receptor-binding-domain (RBD)-based vaccines are considered one of the most effective weapons against COVID-19. During the first step of assessing vaccine immunogenicity, a mouse model is often used. In this paper, we tested the use of five experimental animals (mice, hamsters, rabbits, ferrets, and chickens) for RBD immunogenicity assessments. The humoral immune response was evaluated by ELISA and virus-neutralization assays. The data obtained show hamsters to be the least suitable candidates for RBD immunogenicity testing and, hence, assessing the protective efficacy of RBD-based vaccines.

## 1. Introduction

Over the past two decades, coronaviruses have caused epidemic outbreaks of two respiratory diseases: Middle East respiratory syndrome (MERS) and severe acute respiratory syndrome (SARS). The novel coronavirus disease COVID-19, caused by severe acute respiratory syndrome coronavirus 2 (SARS-CoV-2), which was first detected in China, has reached pandemic levels. To date, a lot of vaccines have been developed to combat COVID-19; many of them are designed to induce antibodies against the receptor-binding domain (RBD) of the SARS-CoV-2 spike protein. The interaction between RBD and the cell surface protein ACE2 plays a key role in the entry of the virus into target cells. The RBD contains a Critical Neutralizing Domain (CND) that could lead to a highly potent neutralizing antibody response as well as cross-protection from multiple strains of SARS-CoV-2. Furthermore, a subunit vaccine based on the RBD is expected to be safer than others [1,2,3].

The first approved RBD-dimer vaccine, ZF2001, is already used in China, Indonesia, and Uzbekistan [4]. At least five more RBD-based vaccines are currently in use in other countries (https://covid19.trackvaccines.org/vaccines/approved/ (accessed on 1 April 2022)).

Despite some successes in vaccination, the development of new effective means of therapy and prophylaxis of COVID-19 remains demand. Therefore, it is of great significance to select appropriate animal models that mimic the biology of human SARS-CoV-2 infection. In February 2020, the World Health Organization (WHO) convened a group of experts (WHO-COM) to develop preclinical models of SARS-CoV-2 infection. The WHO-COM indicated four species as main models for COVID-19: mice, ferrets, hamsters, and non-human primates [5]. 

Non-human primates (NHP), including rhesus macaques (*Macaca mulatta*), cynomolgus macaques (*Macaca fascicularis*), common marmosets (*Callithrix jacchus*), and baboons (*Papio hamadryas*), are more reliable models as they are physiologically, immunologically and genetically more closely related to humans. Recent studies have reported the immunogenicity and protective efficacy of several candidates for a COVID-19 vaccine in the NHP model [6,7,8,9]. The main disadvantage of NHP models is the cost, which limits the number of animals that can be included in the study and therefore negatively affects the statistical results. In addition, most NHPs have a wide variability in genetic background, which sometimes makes it difficult to interpret the results of the study [10].

In contrast, small animal models can be used in large numbers and they and reagents for studying their immune response are readily available [11].

Ferrets (*Mustela putorius furo*) are excellent for studying SARS-CoV-2 transmission and mild infection. They are naturally susceptible to SARS-CoV-2 infection and show mild clinical symptoms similar to humans [12,13,14]. Although ferrets have been used to study the protective properties of vaccines, data on their susceptibility to different isolates or strains of the virus vary between laboratories [15,16,17,18].

Syrian, or Golden, hamsters (*Mesocricetus auratus*) show signs of mild to moderate severity disease with progressive weight loss (1–2 days after infection). Of particular interest is the fact that infection of hamsters with SARS-CoV-2 reflects some of the age and sex differences of COVID-19 in humans. Viral RNA is readily detectable in the respiratory tract and other tissues (e.g., in the small intestine), which may be useful for evaluating therapeutics and vaccines [5]. Because studies in hamsters can be completed quickly and in a cost-effective manner, there is an increasing interest in the use of hamsters for accessing vaccines [19,20,21,22,23,24].

Unlike to those mentioned models, mice (*Mus musculus*) are poorly susceptible to SARS-CoV-2 virus infection because the mouse ACE2 receptor does not effectively bind the viral spike protein [25]. Several strategies have been proposed to address this problem, including virus adaptation and genetically engineered mice [26,27,28]. The use of the mouse model can be explained by the fact that laboratory mice are widely available, well studied, and a wide range of tools are available for studying their immunity. 

To predict the immunogenicity of RBD-based vaccines mice, non-human primate, rabbit, ferret, and horse models have been used [6,7,8,9,10]. At the same time, there are no publications that use hamster models to study RBD-based vaccines. The only exception is the preprint by Wu et al., in which the authors observed a weak humoral immune response to RBD in immunized hamsters [29]. This case and the disturbing lack of hamsters in RBD-based subunit vaccine studies suggests that the predictive value of this animal model may be weak, specifically for vaccines that induce antibodies against RBD.

In this work, we conduct a comparative assessment of the immunogenicity of the SARS-CoV-2 spike RBD in five animal models (mice, hamsters, ferrets, rabbits, and chickens) and try to answer the question of which animal models could be used for vaccine evaluation.

## 2. Materials and Methods

### 2.1. Plasmid Construction, Recombinant Protein Expression, and Purification 

The RBD and S-trimer were prepared as previously described [30]. The Wuhan-1 strain spike nucleotide sequence (GenBank: MN908947) was codon-optimized and synthesized. The RBD region (308 V–542 N) was amplified and cloned into the pVEAL2 transposon plasmid in frame with the N-terminal spike signal sequence (MFVFLVLLPLVSSQC) and the C-terminal 10 × His-tag.

CHO-K1 cells were transfected with the pVEAL2-S-RBD and helper plasmid pCMV (CAT) T7-SB100, encoding SB100 transposase, using Lipofectamine 3000 (Invitrogen, Carlsbad, CA, USA). The transfected cells were selected with puromycin (10 µg/mL), and high-producing clones were obtained by dilution cloning and cultured in roller bottles at 37 °C on DMEM/F-12 (1:1) medium supplemented with 2% FBS and 50 µg/mL gentamicin.

The RBD protein from the CHO-K1 culture medium was purified by Ni-NTA and ion-exchange chromatography and analyzed by SDS-PAGE in a 15% separating polyacrylamide gel. The RBD samples were dialyzed against PBS and sterilized by filtration through 0.22 µm filters. 

The S-protein 1M-P1213 coding gene fragment was designed with a removed protease cleavage site, K986P and V987P amino acid stabilizing substitutions, and a C-terminal T4 bacteriophage fibritin trimerization domain (GYIPEAPRDGQAYVRKDGEWVLLSTFL) followed by a 10 × His-tag. The resulting DNA fragment was cloned into the pVEAL2 vector and produced using the CHO-K1 cell line as described above.

### 2.2. Animal Immunization

All the experimental protocols and procedures were approved by the SRC VB Vector Bioethics Committee (SRC VB Vector/10 September 2020, approved by the protocol of Bioethics Committee No. 5 as of 1 October 2020). 

Groups of BALB/c mice (*n* = 8 animals per group), chinchilla rabbits (*n* = 6), Syrian hamsters (*n* = 10), ferrets (*n* = 6), and Lohmann Brown chickens (*n* = 2) were maintained in separate cages under standard conditions and had free access to water and food at all times. 

Mice (1.5-month-old), hamsters (1-month-old), ferrets (4.5-month-old), and rabbits (2.5-month-old) were immunized intramuscularly twice in a three-week interval with 50 µg of the RBD in combination with Al(OH)_3_. Blood samples were obtained two weeks after immunization, incubated for 1 h at 37 °C and for 2 h at 4 °C, and then centrifuged at 7000× *g* for 10 min. The sera were deactivated by heating for 30 min at 56 °C and stored at −20 °C.

The 3-month-old Lohmann Brown hens, after 4 weeks of adaptive rearing and 2 weeks after laying eggs, began to be immunized. Fifty milligrams of the RBD protein mixed with Al(OH)_3_ was intramuscularly injected into each part of the armpit of chicken wings. Immunization was carried out once every 3 weeks 7 times. All the laid eggs were collected and stored at 4 °C.

### 2.3. IgY Preparation

IgY antibodies were extracted from immunized egg yolk through a precipitation procedure [31]. Briefly, egg yolk separated using an egg separator was mixed with an equal volume of PBS, and then, 3.5% (*v*/*v*) PEG-6000 was added. After incubation at 100 rpm for 10 min, the samples were centrifuged at 4 °C at 4000× *g* for 20 min. The yolk lipids were removed by filtration. Then, 8.5% (*v*/*v*) PEG-6000 was added to the filtrate, and the mixture was incubated at 100 rpm for 10 min. The samples were then centrifuged at 4 °C at 4000× *g* for 20 min, and the sediment was dissolved in 10 mL of PBS. Then, 1.2 g of PEG-6000 was added; the mix was incubated and centrifuged. Sediment dissolved in 1.2 mL of PBS was dialyzed by membrane filtration (MWCO 12–14 kDa) and stored at −20 °C.

### 2.4. SARS-CoV-2 Lysate Preparation

A SARS-CoV-2/human/RUS/Nsk-FRCFTM-1/2020 strain isolated from a patient was obtained from the 48th Central Research Institute of the Ministry of Defense of the Russian Federation. The virus was prepared as previously described [32]. A Vero cell culture monolayer was infected with virus (0.01 TCID50); on day 5, the flask with a total CPE and 100% cell separation from the plastic was subjected to triple freezing/thawing; the virus was purified using sucrose-density-gradient ultracentrifugation and lysed in a lysis buffer (0.25 M Tris–HCl, pH 6.8; 4% (*w*/*v*) sodium dodecyl sulfate; 10% (*v*/*v*) glycerol; 10% (*v*/*v*) β-mercaptoethanol; 0.02% (*w*/*v*) Bromophenol Blue) at a ratio of 1:1. 

### 2.5. ELISA Assays

The 96-well plates (Corning, Glendale, AZ, USA) were coated with 100 µL of antigen (1 µg/mL) in 2 M urea in PBS overnight at 4 °C. The plates were then washed with PBST (PBS containing 0.05% Tween-20) and blocked with a blocking buffer (PBST containing 1% of casein) for 1.5 h at RT. Convalescent and healthy donor sera, immunized animal sera, or IgY preparations were serially diluted and added to the blocked plates. After incubation at RT for 1 h, the plates were washed three times with PBST and incubated with goat anti-human IgG-HRP (GenScript Piscataway, NJ, USA); goat anti-mouse IgG-HRP (Sigma–Aldrich, St. Louis, MO, USA); goat anti-rabbit IgG-HRP (Sigma–Aldrich, St. Louis, MO, USA); anti-hamster IgG-HRP (Invitrogen, Carlsbad, CA, USA); goat anti-ferret IgG, IgA, and IgM-HRP (Sigma-Aldrich, St. Louis, MO, USA); or anti-IgY-HRP (1:20,000) (Thermo Fisher Scientific, Carlsbad, CA, USA) secondary antibodies at RT for 1 h. The plates were washed three times with PBST, and the HRP substrate TMB (Amresco, Solon, OH, USA) was added. The reactions were stopped using 1N HCl. The absorbance was measured at 450 nm using a Varioskan Lux multimode microplate reader (Thermo Fisher Scientific, Waltham, MA, USA). 

The endpoint titer for each serum was defined as the maximum dilution at which a positive result was obtained (>median + 3 × SD of the ODs of the negative controls).

### 2.6. Viral Neutralization Test 

The SARS-CoV-2 sera’s neutralizing antibody titers were determined through cytopathic effect (CPE) inhibition assays.

Vero E6 cells were seeded in 96-well tissue culture plates (Thermo Fisher Scientific, Waltham, MA, USA) and cultured for 24 h to form monolayers. Serial two-fold dilutions of serum samples (1/10 to 1/5120) were mixed with equal volumes of virus suspension containing the 100 TCID50 SARS-CoV-2 coronavirus strain nCoV/Victoria/1/2020 (obtained from the State Collection of Causative Agents of Viral Infections and Rickettsioses SRC VB Vector, Russia). The mixture was incubated at 37 °C for 1 h before adding it to Vero E6 cells. Four days post-infection, the cells were stained with 0.2% gentian violet solution. The presence of a specific cytopathic effect was assessed visually through microscopic examination.

The neutralizing antibody titers were defined as the dilutions of serum that completely prevented the CPE in 50% of the wells.

### 2.7. Statistical Analysis 

All the statistical analyses were performed using the GraphPad Prism 8.0 software, with *p* < 0.05 considered to indicate statistical significance. The statistical significance among different animal groups was determined using a two-tailed nonparametric Mann–Whitney U test with a 95% confidence interval or the Kruskal–Wallis test (for more than 2 groups).

## 3. Results

### 3.1. Recombinant RBD Development and Characterization

The recombinant RBD domain of the SARS-CoV-2 S-protein and recombinant S-trimer were obtained using the CHO-K1 expression system and purified using Ni-NTA affinity chromatography and ion-exchange chromatography as previously described [30]. Purified proteins were analyzed in 10% SDS-PAGE under non-reducing and reducing conditions; as expected, RBD is synthesized in a monomeric form, and S protein forms a trimer when produced in mammalian cells (Figure 1A). Both the RBD and S-trimer reacted well with antibodies in the COVID-19-convalescent human sera in ELISA (Figure 1B).

### 3.2. Immunogenicity of RBD in Mice, Hamsters, Ferrets, and Rabbits

BALB/c mice, Syrian hamsters, ferrets, and chinchilla rabbits were immunized with 50 µg of the RBD in combination with Al(OH)_3_. The immunogen was administrated intramuscularly twice on the 1st and 21st days. No abnormalities in body weight or temperature, or other clinical manifestations were observed in the immunized animals, suggesting a good safety profile for the RBD.

Two weeks after booster prime injection, blood samples were collected from animals and tested through ELISA; the detectable anti-RBD, anti-spike, and anti-SARS-CoV-2 IgG levels were determined (Figure 2A). However, in the hamster group, a wide range (405–32,805) of anti-RBD IgG titers was observed, and titers were low when using the viral lysate as an immunosorbent (Figure 2A). 

The comparative SARS-CoV-2 virus-neutralization analysis of animal sera showed surprising results (Figure 2B). It was found that only 3 out of 10 hamsters had neutralizing antibodies against the SARS-CoV-2 virus (titer ≥ 1/10). The range of titers was 1/10–1/80, which is significantly lower than in other groups of animals.

The Kruskal–Wallis test revealed no significant differences between the mice’s, rabbits’, and ferrets’ neutralizing antibody titers, despite the differentiation in the specific antibody titers observed in the ELISA. 

Since the immune response is dose-dependent, in a separate experiment we examined the effect of the RBD dose on the humoral immune response in hamsters. Animals (4 per group) received 10, 50, or 250 µg of RBD in combination with an aluminum adjuvant twice, with a 14-day interval. The dose of 250 µg of RBD led to increasing specific IgG response in hamsters; however, neutralizing antibody titers were undetectable (titer < 1/10) in three out of four animals (Figure 3). Additionally, there was no statistically significant difference in ELISA between the group that received 10 µg of RBD and the control group.

### 3.3. Dynamic of Humoral Immune Responses in Rabbits and Chickens

Deciphering the dynamic changes in specific antibodies is essential for understanding the immune response to COVID-19 vaccines. Since the manipulation of regular blood sampling in mice and ferrets is laborious, we assessed whether the more-convenient rabbit and chicken models were appropriate for assessing the long-term dynamics of the immune response after immunization. Rabbits were immunized twice with alum-adjuvanted RBD as described above; blood serum samples were obtained every 2 weeks and specific IgG titers were determined by ELISA. The dynamics of the IgG immune responses in the rabbits within four months of prime injection (Figure 4A) suggest that rabbits are a good predictive model for long-term observations. 

We also assessed whether chickens might be a suitable animal model for this purpose. The accumulation of specific IgY antibodies in the egg yolks of chickens does not require blood sampling and can be carried out at a high frequency, depending on the egg-laying. In this paper, two hens received a prime dose of 50 µg of RBD in combination with the aluminum adjuvant and six booster doses of 50 µg of immunogen approximately every 3 weeks. The IgY antibodies were extracted from the egg yolk and tested for reactivity to recombinant RBD via ELISA. The IgY immune response in hens developed slowly, reaching a maximum at 26 weeks after the first injection (Figure 4B).

## 4. Discussion

Before clinical trials, COVID-19 vaccines need pre-clinical evaluations to ensure their safety and efficacy. Animal models are fundamental and essential needs in this phase. In this paper, we evaluated small animal models to assess the immunogenicity of RBD-based COVID-19 vaccines. We tested the immunogenicity of the RBD (50 μg per animal) in combination with an aluminum adjuvant in Syrian hamsters, BALB/c mice, ferrets, and rabbits (Figure 2) and found that only the hamsters developed a weak humoral immune response to the RBD. The geometric mean titer (GMT) of the anti-RBD IgGs in the hamsters’ sera was 0.97 × 10^4^. By contrast, mice, rabbits, and ferrets had comparably high levels of specific antibodies (GMT = 2.9 × 10^6^, 0.4 × 10^6^, and 0.1 × 10^6^, respectively).

The seroconversion rate of neutralizing antibodies in hamsters was only 30% (titers range: 10–40) versus 100% seroconversion in all three groups: GMT = 640 (range: 160–2560) in mice, GMT = 1016 (range: 320–2560) in rabbits, and GMT = 640 (range: 160–1280) in ferrets. The whole specific Ig titers in the ferrets were lower than the IgG titers in the mice’s and rabbits’ sera. Despite this, no statistical significance was observed for the difference between ferrets and rabbits (*p* > 0.29) or ferrets and mice (*p* > 0.9) regarding the virus-neutralizing antibody titers. This shows that the animal model, which is naturally susceptible to infection, probably appropriately reflects the immune response to immunization with the RBD. 

In a separate experiment, RBD-dose-dependent IgG seroconversion in hamsters was observed. Immunization with 250 or 50 μg of RBD elicited specific IgG responses, while no specific antibodies were detected during immunization with 10 μg of the immunogen (Figure 3). At the same time, an increase in the amount of the immunogen did not ensure the appearance of neutralizing antibodies in 100% of the animals. We obtained similar results in an experiment on the immunization of hamsters with a DNA vaccine encoding RBD. In hamsters, anti-RBD antibody titers were very low compared to those found in mice in our earlier study [33]. 

The results obtained raise the question about the preferred use of a hamster model to evaluate any type of COVID-19 vaccine. Wu et al. reported a comparative study of the immunogenicity of the RBD protein using hamster and non-human primate models. The authors observed a weak humoral immune response to 10 μg of trimeric RBD in hamsters, while the titers of specific antibodies in immunized primates were high. When immunized with S-protein, the humoral immune response in both hamsters and primates was equally high [29]. One more example was published by Dalvey et al. The authors obtained virus-like particles (VLPs) containing the engineered RBD variant with increased immunogenicity. BALB/cJ mice and Syrian hamsters were immunized twice with VLPs (2 μg) with an aluminum adjuvant or with both aluminum adjuvant and CpG. The titers of pseudovirus neutralizing antibodies in the groups of hamsters treated with VLPs was lower than in the groups of mice (10^1^–10^3^ vs. 10^3^–10^7^). Following the boost, the hamsters were challenged with SARS-CoV-2 virus and monitored for body weight change and viral titer postchallenge. Animals that received the VLPs with aluminum + CpG recovered in weight faster than the control group; however, the formulation with alum did not result in a significantly different weight change than what was obtained with the placebo [34].

Moreover, several experiments performed by WHO-COM scientists have utilized experimental alum-adjuvanted and formaldehyde-inactivated whole virus vaccines, showing no neutralizing antibodies post-immunization and no protection against challenge in hamsters, but lung cytokines were markedly skewed toward Th2 [35]. Another study reported that hamsters and hACE-2 transgenic mice immunized with an RBD/nucleocapsid fusion protein immunization resulted in high specific antibody titers with undetectable neutralizing antibodies. Additionally, the protection of animals from SARS-CoV-2 infection was provided by a cellular immune response [36]. 

The results of all these studies cannot be explained by the impossibility of producing protective antibodies by the immune system of hamsters since it has been shown that Syrian hamsters inoculated with high and low doses of the SARS-CoV-2 virus (10^5.6^ and 10^3^ PFU) results in protective titers of neutralizing antibodies (range: 640–1280) [37].

In addition, we hypothesized that rabbits and chickens may be convenient and appropriate models for the long-term study of the humoral immune response to RBD. A. Mykytyn et al. demonstrated that rabbits are susceptible to SARS-CoV-2. While the infection is asymptomatic, infectious virus with peak titers corresponding to ~10^3^ TCID50 could be detected up to day seven post inoculation in the nose. The use of young, immunocompetent, and healthy New Zealand White rabbits in this study, however, may not reflect virus shedding and disease in other rabbit breeds or rabbits at different ages [38]. A recent study reported the first evidence of natural SARS-CoV-2 infections among domestic rabbits in France, most likely acquired from a COVID-19-positive owner [39]. However, it is likely that rabbits are quite poorly susceptible to the SARS-CoV-2 infection. 

We observed that the immune response to RBD in rabbits was quite pronounced, and the RBD-specific IgG GMT reached 2.8 × 10^5^ after the second immunization (Figure 4A). The IgG titers persisted for at least 4 months after immunization (GMT = 0.1 × 10^5^ at the 16th week). 

Chickens are not susceptible to Betacoronaviruses, and therefore they can only be used to test the immunogenicity, but not the protective efficacy of the vaccines [18,40]. It was previously reported that the immunization of chickens with inactivated SARS-CoV-2 or S-protein induces specific and neutralizing IgY antibodies with therapeutic potential [41,42,43]. The use of chickens to assess the immunogenicity of the RBD is apparently not entirely appropriate. In our experiment, we observed a low immune response to the introduction of the RBD (Figure 4B). The immune response developed over several months and reached the highest rates only after the seven-fold administration of the immunogen. Our results differ from those obtained by Frumkin L.R. et al. Those authors demonstrated that immunizing chickens several times with 50 µg of non-glycosylated RBD in an oil emulsion induced broad IgY selectivity against all the current variants of concern and showed a favorable safety profile for chronic administration as intranasal drops in rats and humans. Interestingly, the antibodies were ineffective in the Syrian hamster model against a challenge with a titer of 0.8 or 4 × 10^6^ TCID50 of the SARS-CoV-2 virus [44].

Thus, rabbits are attractive for observing the duration of the humoral immune response to SARS-CoV-2 immunogens and could potentially be challenged with coronavirus to assess the protective properties of COVID-19 vaccines.

## 5. Conclusions

This study revealed unequal immune responses to the SARS-CoV-2 spike RBD in different small animal models. Additionally, unexpected data on RBD immunogenicity assessment in hamsters showed that hamsters have a lower humoral immune response to RBD than mice, ferrets, and rabbits. Notably, we observed undetectable levels of neutralizing antibodies in most hamsters. Therefore, hamsters should be used with caution in assessing the protective efficacy of the RBD-based vaccines, and possibly should be replaced by more appropriate model, such as the ferrets.

In addition, we propose rabbits as a potential useful model for the long-term observation of the dynamics of the immune response to RBD. The suitability of the rabbit model to assess the protective properties of vaccines requires further study.

As each animal model has its strengths and limitations, we recommend choosing the optimal animal model with respect to the research questions being addressed. Investigations using two, or even three, animal models may often be necessary to help to draw definitive conclusions.

## Figures and Tables

**Figure 1 viruses-14-01060-f001:**
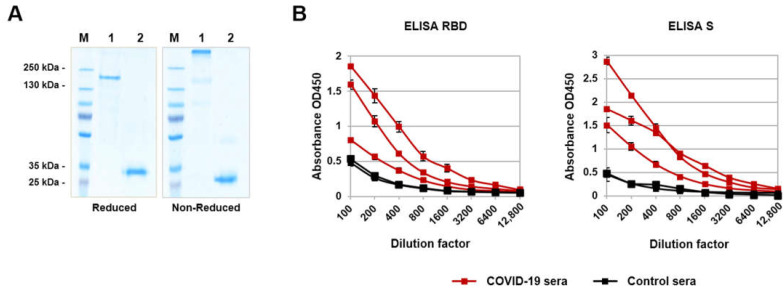
Recombinant RBD and S-trimer characterization: (**A**) reducing and non-reducing 10% SDS-PAGE analyses of purified proteins: M—molecular weight marker; 1—S-trimer; 2—RBD; (**B**) COVID-19-convalescent sera’s reactivity to RBD and S-trimer (mean ± SD). Serum samples of healthy donors were collected before the COVID-19 pandemic, and immune sera were obtained from patients 2–3 weeks after symptom expression and confirmation of the diagnosis by PCR tests for SARS-CoV-2.

**Figure 2 viruses-14-01060-f002:**
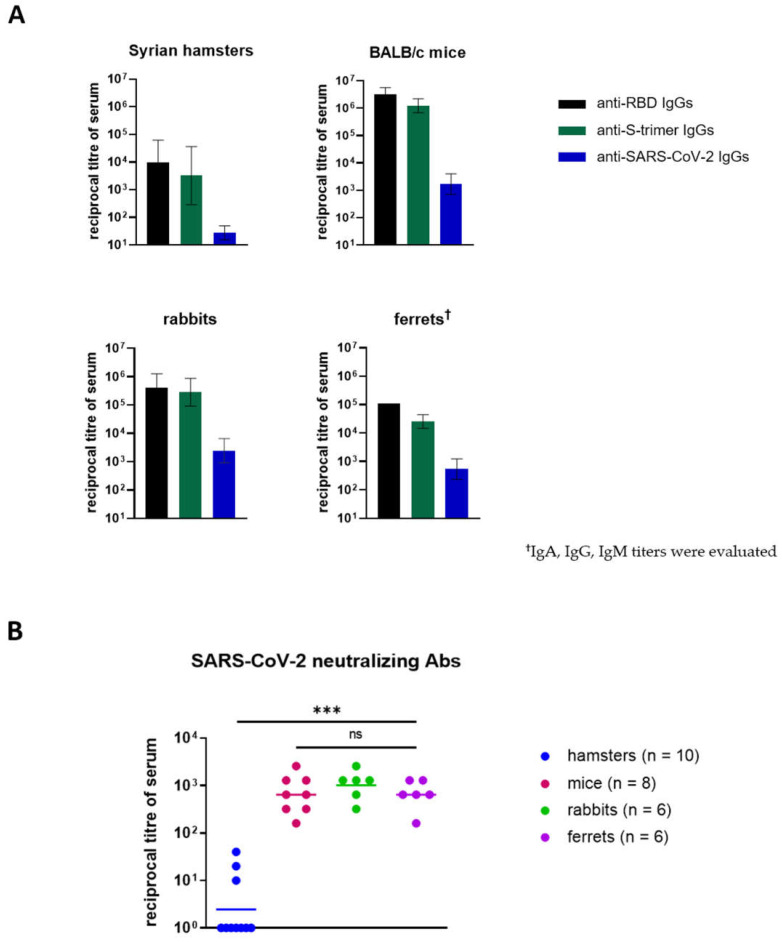
Humoral immune response against RBD protein in animals: (**A**) Animals’ sera’s IgG titers 2 weeks after boost injections according to ELISA using RBD, S-trimer, and SARS-CoV-2 virus lysate (geometric mean ± SD). (**B**) Virus-neutralizing antibody titers evaluated by Victoria/1/2020 SARS-CoV-2 virus-neutralization test (100 TCID50). Horizontal lines represent geometric mean values; Statistical significance was calculated using the Kruskal–Wallis test (*** *p* < 0.001; ns, not significant (*p* > 0.05)). Negative samples (titer < 1/10) were taken as 1.

**Figure 3 viruses-14-01060-f003:**
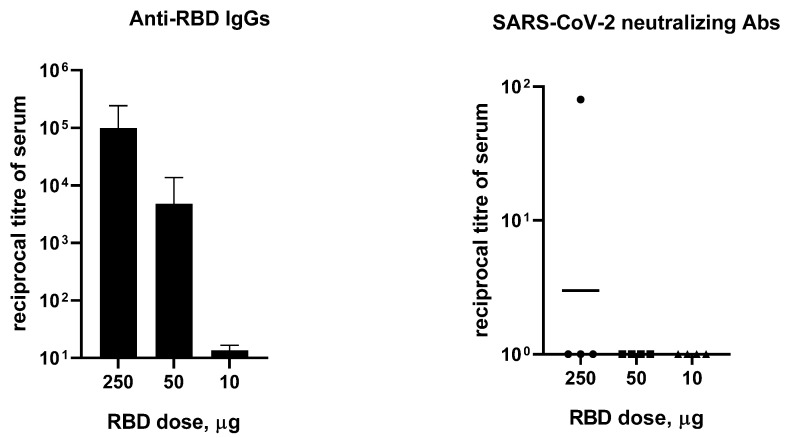
Effect of the dose of RBD protein on hamsters’ (*n* = 4) specific IgG and neutralizing antibody response evaluated through ELISA and Victoria/1/2020 SARS-CoV-2 virus-neutralization tests (100 TCID50). Data presented are geometric mean ± SD. Negative samples (titer < 1/10) were taken as 1.

**Figure 4 viruses-14-01060-f004:**
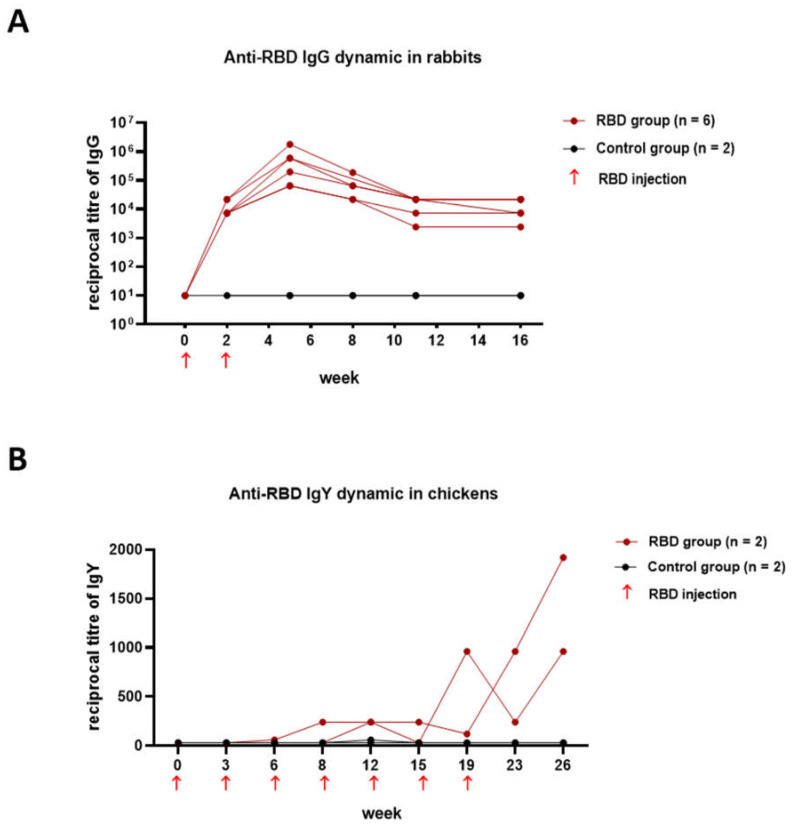
RBD-specific antibodies in the rabbits and chickens evaluated by ELISA (individual curves): (**A**) IgG antibody response in rabbits’ sera; (**B**) IgY antibody response in chickens.

## Data Availability

Not applicable.

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
