# Peer review of "Are Hamsters a Suitable Model for Evaluating the Immunogenicity of RBD-Based Anti-COVID-19 Subunit Vaccines?"

_viruses, 2022, doi:10.3390/v14051060_

Round 1
Reviewer 1 Report
Merkuleva et. al,
The authors assessed the immunogenicity of SARS-Covid-19 RBD by using five experimental animals models mice, hamsters, rabbits, ferrets, and chickens. The authors concluded that the hamsters are the least suitable candidates for RBD immunogenicity testing. In fact, to find the best animal model to evaluate the RBD immunogenicity, the animal models should show the most closed immune response to humans and be able to perform the challenging study. The information will be useful information for evaluating the Covid-19 vaccine efficacy. In addition, important positive control, such as human vaccines should be included. In addition, there are some comments listed below:
1. Did the authors treat the sera with Receptor Destroy Enzyme (RDE) to remove the non-specific binding?
2. The authors should show the data of the reaction of S protein with human sera.
3. In figure 1B, the authors should explain the sera indicating the different colors? Which sera were collected before the Covid-19 pandemic and which sera were collected from Covid-19 patients?
4, The animals were immunized with 50 µg RBD protein. The dosage is very high for the small animal model. How did the authors determine this dosage?
5. In the Introduction section, the hamster models should be used for assessing the immunogenicity of Covid vaccines. The authors should cite those papers.
6. In lines 245-258, the conclusion is not making sense.
Author Response
Thank you for your comments, which have helped us to greatly improve the manuscript. Below we have provided brief explanations and answers to Your questions.
1) Did the authors treat the sera with Receptor Destroy Enzyme (RDE) to remove the non-specific binding?
The Receptor Destroy Enzyme is commonly used in the hemagglutination inhibition (HI) assay to eliminate non-specific hemagglutination inhibitors existing in a serum specimen.
In our study, we did not use this enzyme. The test of inhibition of the cytopathic effect (CPE) of a virus in cells does not usually require an enzyme. Tests were carried out in compliance with all standards. We, of course, tested sera from non-immunized animals as controls and did not observe non-specific inhibition of the virus.
2) The authors should show the data of the reaction of S protein with human sera.
We have added the S-protein reaction with human blood sera to the article to confirm the antigenicity of the obtained protein.
3) In figure 1B, the authors should explain the sera indicating the different colors? Which sera were collected before the Covid-19 pandemic and which sera were collected from Covid-19 patients?
We agree with your remark. We took this into account and tried to make the figure 1B more clear.
4) The animals were immunized with 50 µg RBD protein. The dosage is very high for the small animal model. How did the authors determine this dosage?
To determine the dose of the immunogen, we analyzed the literature data. We have noted, for small animals it is acceptable to repeatedly increase the dose of the immunogen up to several hundred micrograms (e.g.100 µg per mouse, doi: 10.1101/pdb.prot100297).
We focused on a dose of 50 µg of RBD because it is used in approved human vaccines. The doses were chosen, among other things, based on our preliminary studies in mice and hamsters. We hypothesized that smaller doses of the immunogen may cause a weak immune response in some animal models, and the use of a somewhat excessive amount of the immunogen would help to avoid this. In addition, we wanted to test not only immunogenisuty the safety profile of the protein in these animals.
5) In the Introduction section, the hamster models should be used for assessing the immunogenicity of Covid vaccines. The authors should cite those papers.
We think that the introduction and discussion sections were poorly written. We have added information from the literature to improve the manuscript.
6) In lines 245-258, the conclusion is not making sense.
We agree with you that this experiment does not make sense in this work. It has been removed from the manuscript.
«In addition, important positive control, such as human vaccines should be included».
In our study, we evaluated the immune response to RBD in several popular small animal models. RBD contains the main epitopes of neutralizing antibodies, therefore, it is assumed that antibodies to this domain play a major role in protection after vaccination or past infection.
We agree with you that it would be better to add approved RBD-based vaccines as controls. At the time of the study, only the RBD-dimer vaccine, ZF2001, developed in China was known, and unfortunately it was not available in our country.
However, we hope that our modest study will be useful for researchers developing vaccines based on RBD.
Author Response
Thank you for your comments, which have helped us to greatly improve the manuscript. Below we have provided brief explanations and answers to Your questions.
We have tried to correct the logic of presentation and present the results more clearly. We also agree with all Your comments and have made appropriate changes to the text of the manuscript.
Line 73. Please define the abbreviation of a.a.
We have replaced it with a more concise phrase «RBD region (308V–542N)»
Line 86. T4 bacteriophage fibritin trimerization domain. Please cite a proper reference or list the a.a. sequence.
Amino acid sequence of T4 bacteriophage fibritin trimerization domain has been added to the manuscript.
Line 113. What’s the age of the animals?
The age of the animals has been added to the text.
Line 129. What does thrice mean?
We have replaced «Thricle» with the correct translation («three times»).
Line 146. Please define CPE/cell? Is it similar to TICD50?
You are right, we meant TICD50. The mistake has been corrected
Line 175 Please correct the TCID50.
All TCID50 have been corrected.
Line 199-203. What did the colored lines represent?
We have tried to make the figures more clear
Line 251. Please define VNT, and explain why use 69 TCID50?
+
Line 254-258. Please add several sentences about why the authors did this experiment.
We think that this experiment does not make sense in this work. It has been removed from the manuscript.
Line 273-275. There are 2 and 4 red arrows in Figures 4A and 4B, respectively. I guess the arrow indicates the immunized date; however, why not the arrow marked the XAxis? It’s confused to guess the meaning of the arrows. Another problem is that? What dose the colored lines mean? The authors only illustrate the control group. In addition, the authors should briefly describe how the rabbits and chickens were immunized.
We have corrected the figures and hope that now they will be clear to the reader. We have added description of the rabbits and chickens immunization.

Reviewer 3 Report
In this study, Iuliia A. Merkuleva et al. evaluated the immunogenicity of SARS-CoV-2 RBD using five different animal models. They found that the monomeric RBD induced fewer specific antibodies and almost no neutralizing antibodies in hamster than did in other animals. And rabbits represent a compromise option for assessing the duration of the humoral response. The study provides some implications for the selection of animal models for subunit vaccine assessment, especially the RBD subunit vaccine.
While the different animal models have different weight, all animals received 50ug protein for immunization. Could the authors explain this? If dosage is based on body weight, hamster will receive more RBD. Higher dosage for hamsters seems to induce more antibodies as shown in figure3A.
Figure3A The effect of RBD dosage on hamsters’ specific IgG response was evaluated. Was the neutralizing activity of the serum from different dosage tested? It’s better to show if higher IgG response also gives higher neutralizing titer.
Figure4 Please explain in the figure legends if the curves with different colors mean different samples from the same group. And is there only one animal for the control group?
In general, to evaluate a vaccine in an animal model. One of the important parts is to study the protection of the animals from virus challenge. So are rabbits and hens susceptible to SARS-VoV-2? If not, it will be hard to evaluate whether the induced antibodies by the vaccine in the animal models can provide good protection.
Author Response
Thank you for your comments, which have helped us to greatly improve the manuscript. Below we have provided brief explanations and answers to Your questions.
While the different animal models have different weight, all animals received 50 ug protein for immunization. Could the authors explain this? If dosage is based on body weight, hamster will receive more RBD. Higher dosage for hamsters seems to induce more antibodies as shown in figure3A.
Unfortunately, there are currently no strict recommendations on vaccine doses. Calculating the dose in relation to the weight of the animal does not guarantee a good immune response, although this is done when testing therapeutic agents (eg immunoglobulins). Differences in the immune system of animals can lead to unexpected results. Small animals may respond strongly to higher doses of the immunogen than larger animals.
The doses were chosen, among other things, based on our preliminary studies in mice and hamsters.
Figure3A The effect of RBD dosage on hamsters’ specific IgG response was evaluated. Was the neutralizing activity of the serum from different dosage tested? It’s better to show if higher IgG response also gives higher neutralizing titer.
We evaluated the titer of neutralizing antibodies to RBD doses. These data have been added to the manuscript.
Figure4 Please explain in the figure legends if the curves with different colors mean different samples from the same group. And is there only one animal for the control group?
We have corrected the figures and hope that now they will be clear to the reader.
In general, to evaluate a vaccine in an animal model. One of the important parts is to study the protection of the animals from virus challenge. So are rabbits and hens susceptible to SARS-VoV-2? If not, it will be hard to evaluate whether the induced antibodies by the vaccine in the animal models can provide good protection.
It is likely that readers may also have such a question, so we have added a few sentences on this topic to the manuscript.

Round 2
Reviewer 1 Report
The revised manuscript is much approved. However, the figure 2B is missing
Reviewer 3 Report
My questions have been addressed. Figure2B seems not visible.